# Effects of MgO Expansive Agent and Steel Fiber on Crack Resistance of a Bridge Deck

**DOI:** 10.3390/ma13143074

**Published:** 2020-07-09

**Authors:** Feifei Jiang, Min Deng, Liwu Mo, Wenqing Wu

**Affiliations:** 1College of Materials Science and Engineering, Nanjing Tech University, Nanjing 211800, China; dengmin@njtech.edu.cn (M.D.); andymoliwu@njtech.edu.cn (L.M.); 2College of Naval Architecture Civil Engineering, Jiangsu University of Science and Technology, Zhangjiagang Campus, Suzhou 215600, China; 3School of Transportation, Southeast University, Nanjing 210089, China; wuwenqing@seu.edu.cn

**Keywords:** MgO expansive agent, steel fiber reinforced concrete, bridge deck, laminated slab, shrinkage crack

## Abstract

To prevent cracks caused by shrinkage of the deck of the Xiaoqing River Bridge, MgO concrete (MC) and steel fiber reinforced MgO concrete (SMC) were used. The deformation and strength of the deck were measured in the field, the resistance to chloride penetration of the concrete was measured in the laboratory, and the pore structure of the concrete was analyzed by a mercury intrusion porosimeter (MIP). The results showed that the expansion caused by the hydration of MgO could suppress the shrinkage of the bridge deck, and the deformation of the deck changed from −88.3 × 10^−6^ to 24.9 × 10^−6^, effectively preventing shrinkage cracks. At the same time, due to the restriction of the expansion of MgO by the steel bars, the expansion of the bridge deck in the later stage gradually stabilized, and no harmful expansion was produced. When steel fiber and MgO were used at the same time, the three-dimensional distribution of steel fiber further limited the expansion of MgO. The hydration expansion of MgO in confined space reduced the porosity of concrete, optimized the pore structure, and improved the strength and durability of concrete. The research on the performance of concrete in the in-situ test section showed that MgO and steel fiber were safe for the bridge deck, which not only solved the problem of shrinkage cracking of the bridge deck but also further improved the mechanical properties of the bridge deck.

## 1. Introduction

Cracks are the direct cause of concrete penetration, which seriously affect the durability of concrete. Due to shrinkage and temperature changes, concrete volume will be deformed, which is the main reason for early cracks. From the perspective of structural calculation, deformation under free conditions will not produce tensile stress, and the concrete will not crack. However, once the deformation is restrained, tensile stress will be generated, and since the tensile strength of concrete is very low, it is easy to crack. In thin-walled structures such as bridge decks, due to the large surface area, the volume deformation caused by shrinkage and temperature is greater, and the cracking caused by the constraints of steel bars and bearings is also more serious. [1,2]. Early deformation cracks do not necessarily affect the safety of the structure immediately, but will seriously affect the durability of the bridge, so it is necessary to pay attention to the damage caused by cracks [3,4,5].

The cracks of cement-based materials are closely related to the non-load deformation of the materials. American academic Bazant pointed out that the early deformation of concrete can be divided into drying shrinkage, chemical shrinkage, and thermal shrinkage [6]. Early non-load cracks easily make concrete more vulnerable to the invasion of potentially harmful substances, and this kind of crack is more serious in bridge engineering for three main reasons. The first reason is that the concrete for bridges has high strength and low water-to-binder ratio (w/b), so it shrinks greatly [7,8,9]. It is reported that chemical shrinkage occurs in concrete when the w/b ratio is less than about 0.42, and increases with the decrease of w/b [10]. In addition, the bridge deck with a thin-walled structure has a large contact area with air, resulting in large drying shrinkage. Furthermore, due to direct sunlight, the moisture loss of the concrete of the bridge deck is quicker, which also aggravates the drying shrinkage. The second reason is that, compared to the concrete of houses, there are more steel bars in bridges. These steel bars severely restrict shrinkage and produce greater tensile stress. The third reason is large temperature cracks. Bridges are in field environments without any other shelter from buildings. The temperature affected by solar radiation is high in the daytime, reaching 40–50 °C, but at night, due to the small thickness of the bridge deck, the heat will be quickly distributed to the air, and the temperature will be rapidly reduced to 20 °C. The strength of fresh concrete is low, and temperature cracks easily occur with such large temperature differences.

The bridge deck is an important part of bridge. It not only needs to have high strength to bear automobile loads but also needs to have good crack resistance and waterproof performance to ensure that the rainwater will not penetrate into concrete and lead to corrosion of steel bars [11,12,13]. For example, only two months after the Shantou Bay Bridge was put into use in China, a large number of cracks appeared on the bridge deck. What is more, two years later, the cracking area reached 75% of the total area of the deck, causing great economic losses. This kind of accident reminds researchers of the need to develop new high-performance concrete to reduce shrinkage and improve the crack resistance of bridge decks [14,15,16,17].

The Changshen Expressway is one of the “three vertical” lines of China’s highway network, connecting the Northeast, the Yellow River Delta, the Yangtze River Delta, and the Pearl River Delta and is a major north–south corridor through eastern China. As shown in Figure 1, the Xiaoqing River Bridge is a steel–concrete composite beam bridge on the Changshen Expressway, with a total length of 260 m (70 m + 120 m + 70 m). As shown in Figure 2, the bridge deck is a laminated slab structure. The lower part is a precast concrete slab with a thickness of 80 mm, and the upper part is cast-in-place concrete with a thickness of 240 mm.

In general, the precast concrete slab has been stacked in the construction site for about one month, the shrinkage has been almost completed, and there will be no obvious shrinkage in the later stage. However, the cast-in-place concrete in the upper part will shrink, and the shrinkage is constrained by the precast slabs below, resulting in tensile stress [18,19,20,21]. When the tensile stress exceeds the tensile strength, cracks will be produced in the deck [22].

Shrinkage of concrete, as a material property, has been extensively studied in the past decade, and researchers have proposed many measures to reduce shrinkage cracks. Huang added 8% MgO to the concrete, which caused the concrete to expand by 60 × 10^−6^, preventing shrinkage cracks in airport pavement. However, at the same time, MgO reduced the compressive strength of concrete by 27% [23]. Yousefieh studied the effect of fibers on drying shrinkage and crack resistance of concrete, and he found that steel fibers could reduce the width and length of cracks [24]. From the above research, we can find that when MgO is used alone, the expansion of MgO can compensate the shrinkage of the concrete and prevent cracking, but it reduces the strength. When steel fiber is used alone, the steel fiber can only prevent the further expansion of cracks after cracks occur, but cannot prevent the generation of cracks. In order to prevent shrinkage cracks without reducing the strength of concrete, MgO and steel fibers are used simultaneously in this research.

MgO has primarily been used in dam and airport pavement to reduce shrinkage cracks. Many researchers have proved that MgO concrete is superior to ordinary concrete in terms of carbonization resistance, chloride ion penetration resistance, sulfate corrosion resistance, and freeze-thaw cycle, which means that MgO can also be used in bridges to achieve the purpose of inhibiting shrinkage cracks [25,26,27,28,29,30,31]. In the previous study, our research team invented steel fiber reinforced MgO concrete (SMC) by using both MgO and steel fiber [32]. In addition, the deformation of SMC in the laboratory environment was studied, and it was found that SMC can produce micro expansion and prevent early shrinkage cracks. However, this kind of new material has remained in laboratory research and has not been applied in practical engineering. In the laboratory environment, the temperature and humidity have remained stable, but in the field environment, the temperature and humidity are constantly changing, and even the temperature change within a day reaches 20 °C. In the field environment, the deformation and strength of SMC have not been tested, which hinders the application and promotion of SMC. In this research, SMC was used in the cast-in-place deck of the Xiaoqing River Bridge, and the effect of SMC was evaluated by analyzing temperature, deformation, and strength. The study proved that this specially designed concrete of the bridge deck was able to maintain good performance under the harsh field environment. In addition, the deck is required to have good durability, because it not only bears vehicle loads but also is eroded by rain water. In this study, the durability of the bridge deck was measured by chloride permeability. Thanks to the positive results of this study, the combined application of MgO and steel fiber in concrete should be highly promoted for building bridge decks of long span bridges.

## 2. Materials and Methods

### 2.1. Materials

In this research, the cement was Class 52.5 ordinary Portland cement (Shan Aluminum Cement Co., Ltd., Shandong, China). Fly ash was Class I fly ash (Shenhua Guohua Shoudian Electric Co., Ltd., Beijing, China). The chemical composition of cement and fly ash was tested according to GBT 176, 2017 “Chemical Analysis Method of Cement” [33], and the test results are shown in Table 1. The coarse aggregate was limestone macadam with 5–20 mm continuous grading. Its apparent density was 2.73 g/cm^3^, its bulk density was 1.48 g/cm^3^, its porosity was 46.7%, and its water absorption was 0.45%. The fine aggregate was river sand, with a fineness modulus of 2.94, apparent density of 2.63 g/cm^3^, bulk density of 1.54 g/cm^3^, porosity of 41.7%, and water absorption of 0.40%. The MgO expansion agent was produced by Wuhan Sanyuan Special Building Materials Co. LTD in China, and its activity was 115 s. The steel fiber was wave shaped steel fiber (SF, Shuanglian Building Materials Co., Ltd., Shandong, China), and its detailed parameters are shown in Table 2. The mix proportions are listed in Table 3.

### 2.2. Experimental

#### 2.2.1. Temperature and Deformation Test

As shown in Figure 3, a VWS-10 vibrating wire strain gauge was used to measure the temperature and deformation of concrete. The temperature measurement range of the strain gauge was from −40 °C to 80 °C with an accuracy of 0.5 °C. *F* was changed with the change of concrete length, and the deformation of concrete could be calculated by calculating the change value of *F* [34]. The strain was calculated by Formula (1) as follows:(1)ε=kΔF+(b−a)ΔT
where *ε* is expansion or contraction of concrete, in 10^−6^; *k* is the constant coefficient of the strain gauge, in 10^−6^/F; Δ*F* is the change of *F*; *b* is the temperature correction factor, in 10^−6^/°C; *a* is the thermal expansion coefficient of concrete, in 10^−6^/°C; and Δ*T* is the change of the temperature of concrete, in °C.

As shown in Figure 4, in order to compare the effects of different MgO and steel fiber on the performance of the bridge deck, the deck was divided into two parts. The area of the first part was 9 m^2^. The material used was SMC, and S3 was used to measure its temperature and deformation. The area of the second part was 519 m^2^. The material used was MC, and S4 was used to measure the temperature and deformation of MC.

#### 2.2.2. Strength Test

The strength of concrete was tested by a concrete rebound instrument (Gantan Instrument Co., Ltd., Shanghai, China). As shown in Figure 5, in order to evaluate the influence of MgO and steel fiber on the strength, the strength of the bridge deck was tested at 3 days, 7 days, and 21 days. To ensure the accuracy of the experiment, the partial deck was polished before the test. In order to ensure the effectiveness of the rebound strength, before the test the rebound meter was calibrated in accordance with the “Technical Specification for Testing the Compressive Strength of Concrete by the Rebound Method” JTJ/T 23-2011 [35]. The bouncing rod rotated in four directions, each rotation being 90°, and the rebound values were 80.6, 80.5, 80.4, and 80.5, respectively, which met the requirements of the range of 80 ± 2 in the specifications.

#### 2.2.3. Permeability Test

Considering that in winter, in order to prevent the deck from freezing, deicing salt is often sprinkled on the deck. When chloride ions diffuse into the bridge deck, they cause corrosion and expansion of steel bars, which lead to cracking and affect the safety of the structure. In order to analyze the chloride permeability of SMC, a rapid chloride permeability test was carried out according to the procedure described in ASTM C1202 [36,37,38,39]. The test device is shown in Figure 6.

A cylindrical concrete specimen with a height of 50 mm and a length of 100 mm was used to test the chloride diffusion coefficient. Two different solutions, namely 3% NaCl solution and 0.3 mol/L NaOH solution, were selected in the experiment. A concrete chloride flow meter was used to measure the flux through the sample within the specified time. The total electric flux of the concrete test block for 6 h was calculated as follows according to Formula (2):(2)$$Q=900\times \left(I_{0}+2I_{30}+2I_{60}+\cdots +2I_{t}+\cdots +2I_{300}+2I_{330}+I_{360}\right)$$
where *Q* is total flux through the specimen within 6 h (C); *I_0_* is initial flux (A), to 0.001A; *I_t_* is flux (A) at specified time t (min), to 0.001A.

According to the Nernst–Planck equation, the chloride diffusion coefficient (CDC) can be calculated by Q according to Formula (3) as follows:(3)CDC=2.57765+0.00492×Q

#### 2.2.4. Pore Structure Test

The influence of MgO and steel fiber on the pore structure of concrete was analyzed by mercury intrusion porosimeter (MIP). First, the concrete was broken into small pieces with a length of 2 mm, and the coarse aggregate was removed. Then in order to stop the hydration of cement and MgO, the samples were immersed in anhydrous ethanol for 24 h. After that, the samples were vacuum dried for another 12 h. By comparing the changes of the pore structure before and after adding MgO and steel fiber into the concrete, the reasons why the deformation, strength, and durability of bridge deck changed can be explained from the microscopic point of view.

## 3. Results

### 3.1. Temperature Variation

Figure 7 shows the temperature variation on the surface and interior of deck. At the beginning of pouring, the external environment temperature was 25.2 °C, and the data collection was started four hours after pouring. It can be seen from Figure 7 that due to the solar radiation on the surface of the deck, the temperature changed greatly in the daytime and evening, and the biggest range of temperature in a day increased from 20.8 °C to 46.9 °C. On the other hand, due to the poor thermal conductivity of concrete, there was little temperature change inside deck, controlled within 6 °C [40,41,42].

The internal temperature variation of deck could be divided into three stages. The first stage was the rapid temperature rise stage. Due to the exothermic hydration of cement and MgO, the temperature increased rapidly to 40.9 °C at 17 h and reached 46.9 °C at 41 h. On the other hand, the heat on the concrete surface dissipated rapidly, and the temperature dropped rapidly to 37.9 °C, resulting in a temperature difference of 13 °C between the inside and outside of the bridge deck. According to the stress calculation formula (Formula (4)) proposed by Bradbury, the tensile stress increases with the increase of temperature difference [43]. Therefore, the bridge deck easily produced temperature cracks in the early stage when the concrete strength was low. However, the addition of steel fibers significantly reduced cracks due to two main factors. On the one hand, steel fibers increased the thermal conductivity of concrete and reduced the temperature difference; on the other hand, the combined effects of steel fibers and MgO improved the strength of concrete and limited the development of cracks.
(4)$$\sigma_{t}=\frac{1}{2}\Delta T\times E\alpha \times \left(\frac{C_{x}+\mu C_{y}}{1-\mu^{2}}\right)$$
where Δ*T* is the temperature difference between the inside and the surface of the bridge deck (°C); *E* is the elastic modulus of concrete (MPa); *α* is the thermal expansion coefficient of concrete (°C^−1^); *μ* is Poisson’s ratio of concrete; and *C_x_* and *C_y_* are the stress coefficients.

The second stage was the cooling stage. In this stage, the hydration rate slowed down noticeably. As the heat diffused into the air, the internal temperature decreased gradually, and it decreased to 28.1 °C at 106 h. The third stage was the natural temperature stage. After 106 h, most of the hydration of cement and MgO was completed, and the temperature changed with the external environment temperature. Compared with the three different stages, the following conclusions can be reached: (1) In the temperature rise stage, the temperature difference between the inside and outside of the deck was very small, and the concrete was not fully hardened during this time, so it was not easy to produce cracks. (2) In the temperature drop stage, the surface temperature was higher than the internal temperature in the daytime. However at night, the surface temperature was less than the internal temperature. The alternating action of plus–minus temperature stress made it easy to produce temperature cracks in the deck. (3) What is more, in the early stage, the strength of the concrete was low, and the large temperature stress made it easy to produce through-cracks in the bridge deck. Therefore, the maintenance of concrete needed to be strengthened in the temperature drop stage. As shown in Figure 8a, when the membrane curing method was used, it could not only ensure the early hydration of cement, but it also ensured the water demand during the hydration of MgO, which made the early strength of MgO concrete develop stably and improve the early performance of MgO concrete. As shown in Figure 8b, during the construction of the Xiaoqing River Bridge, after 17 h of pouring, the surface of the bridge deck was covered with geotextile, and water was sprayed every 6 h. After 21 days, the bridge deck surface was intact and there were no temperature cracks.

### 3.2. Volume Deformation

Figure 9 shows the change of deformation of concrete with different proportions over time. The deformation of concrete changed periodically with the temperature change of the external environment. Similar to the law of temperature change, deformation can be divided into three stages. The first stage was rapid expansion stage, lasting for 21 h. The REF also expanded at this stage, but its expansion value was smaller than that of the other two kinds of concrete, which indicated that MgO had begun to hydrate and had produced effective expansion at the early stage of temperature rise. For example, at 21 h, the expansion of REF was 48.4 × 10^−6^, which was only 34.9% and 45.2% of MC and SMC, respectively. The second stage was the shrinkage stage, lasting for 100 h. The third stage was the stable stage, at which point the shrinkage or expansion no longer changed dramatically.

It can be seen from Figure 9 that the deformation of REF decreased with time due to drying shrinkage and chemical shrinkage, and the shrinkage reached −88.3 × 10^−^^6^ at 520 h. In this stage, if no measures are taken to reduce shrinkage, the shrinkage of the upper cast-in-place slab will be restrained by the lower precast slab, resulting in tensile stress. When the tensile stress exceeds the tensile strength of the concrete, it will lead to cracking, which was also one of the main reasons for the cracking of many bridge decks in the past. According to Formula (5), the width of the crack increases with the length of concrete slab and the shrinkage deformation [44]. Moreover, in this project, there was no expansion joint in the 44 m long bridge deck. Therefore, if the shrinkage of the bridge deck is not limited, there will be wide cracks in such a long concrete slab.
(5)d=C×L×(ΔTα+ε)
where *d* is the width of the crack; *C* is the constant coefficient; *L* is the length of the concrete slab; Δ*T* is the change in temperature; *α* is the thermal expansion coefficient of concrete; and *ε* is the shrinkage deformation of concrete.

For MC, the shrinkage of concrete was compensated by the expansion of MgO hydration, and the deformation was 24.9 × 10^−^^6^ at 520 h, attributed to the generation of Mg(OH)_2_. MgO made the concrete from original shrinkage to expansion and fundamentally prevented the shrinkage cracking of the bridge deck. For SMC, by contrast, due to the steel fiber restricted the expansion of MgO, part of the energy generated by MgO was used to tension the steel fiber and generated self-stress, which reduced the porosity of concrete and improve the strength of concrete [45,46,47,48]. Furthermore, the expansion of SMC was smaller than that of MC, which was 9.1 × 10^−6^ at 520 h. During the whole time, both MC and SMC were always in the expansion state, so there was no shrinkage cracking, which proved that MgO and steel fiber could be used to solve the early cracks of decks.

### 3.3. Expansion Caused by MgO Hydration

The deformation measured by strain gauge was the comprehensive result of concrete shrinkage and MgO expansion, which was the sum of shrinkage (ε_c_) and expansion (ε_M_). As shown in Figure 10, in order to analyze the influence of steel fiber constraint on MgO expansion, the deformation of MC and SMC simultaneously subtracted the deformation of REF so as to obtain the deformation caused by MgO hydration. The expansion process of MgO could be divided into two stages. The first stage was the rapid expansion stage. In this stage, the expansion caused by rapid hydration of MgO was obviously greater than the shrinkage of concrete. The second stage was the stable stage. In this stage, the hydration rate of MgO slowed down significantly, the expansion and shrinkage were approximately the same, and the total deformation remained stable. By comparing the different expansions of MC and SMC, the following conclusions can be found: (1) In the early days, for MC, the expansion of MgO was able to produce an expansion of 60.7 × 10^−6^, which completely compensated the shrinkage of the concrete. In the later period, the expansion did not increase without limit, but gradually tended to a stable value. Therefore, the use of MgO in the bridge deck is safe and will not cause stability problems due to harmful expansion. (2) Due to the restraint of steel fiber, the expansion of SMC was smaller than that of MC. In the early stage (21 h), the expansion of SMC was 60.7 × 10^−6^, only 65.8% of MC. In the later stage (520 h), the expansion of SMC was 55.8 μ ε, only 48.9% of MC. In contrast, for SMC, the energy generated by the hydration of MgO was not completely used to expand the concrete, but was partially used to tension the steel fibers, which was helpful to improve the density of the concrete.

### 3.4. Strength Test

Figure 11 shows the strength variation of concrete with different proportions. With the increase of curing age, the strength of all concrete gradually increased. Compared with REF, the strength of MC increased by −0.4%, 1.2%, and 1.7% at 3 d, 7 d, and 21 d, respectively, which means that MgO had little effect on the strength of deck. On the other hand, the strength of SMC increased significantly. Compared with REF, the strength of SMC increased by 2.1%, 10.3%, and 17.5% at 3 d, 7 d, and 21 d, respectively. The test results showed that the combined action of MgO and steel fiber reduced the internal defects of concrete and improved the density of concrete, which improved the strength of the deck. The results of the two tests, strength and deformation, showed that the combination of MgO and steel fiber not only prevented the shrinkage crack of the deck, but also improved the strength of the deck, which provided a new method for preventing the early cracks in the deck.

The strength test results showed that after 8% MgO was added to the bridge deck, the strength of the deck at 21 d was improved due to the increase of the total amount of cementing material. However, in the early days, the strength of the matrix was small, and the constraint on expansion of MgO was small. The hydration of MgO produced Mg(OH)_2_, which caused the concrete to expand outward, which increased the porosity and reduced the early strength. Therefore, at 3 d, the strength of MC was 0.4% lower than that of REF. On the other hand, when MgO and steel fiber were used at the same time, the steel fiber restricted the expansion of MgO, and Mg(OH)_2_ grew in the restricted space, which reduced the porosity of the concrete, improved the density, and significantly increased the strength.

### 3.5. Chloride Permeability of Concrete

The results of the rapid chloride permeability test of concrete at different ages are shown in Table 4. It can be seen that MgO and steel fibers enhanced the resistance to chloride penetration of concrete, and the concrete with both MgO and steel fibers had a higher resistance than that of MgO alone.

It can be seen from Table 4 that MgO not only did not reduce the chloride penetration resistance of concrete, but it also improved the chloride penetration resistance of concrete. The increase within 60 days was 4.6%–0.9%, which showed that it was safe to use MgO in the bridge deck. When MgO and steel fiber were used at the same time, the resistance to chloride penetration was further improved, the increase within 60 days was from 12.2% to 40.7%, indicating that the combined effect of MgO and steel fiber made the concrete more dense, and the same conclusion can be obtained in the pore structure test. Therefore, in summary, it was found that MgO could be used alone to suppress the shrinkage cracks of bridge decks in the southern regions with high winter temperatures, and MgO and steel fibers could be used to suppress the shrinkage cracks of bridge decks in the northern regions with low winter temperatures.

### 3.6. Performance Enhancement Mechanism of Concrete

#### 3.6.1. The Relationship between Deformation and Stress

Figure 12 shows the stresses due to the deformation of three types of concrete under the constraint conditions. Under the restraint of reinforcement and precast slab, tensile stress is generated due to shrinkage in REF, and the bridge deck will crack when the tensile stress exceeds the tensile strength of concrete (Figure 12a). In contrast, for MC and SMC, the expansion of MgO compensates for the contracting of the concrete, causing a slight expansion of the bridge deck. Under restraint conditions, compressive stress is generated due to expansion inside the concrete, which not only prevents cracking, but also makes the concrete denser and improves the strength of the concrete (Figure 12b). Furthermore, the greater the constraint, the greater the compressive stress and the denser the concrete. For SMC, under the combined constraint of steel fiber and reinforcement, the concrete produced greater compressive stress, which also explains why the strength of SMC was greater than that of REF and MC.

#### 3.6.2. Pore Structure of Concrete

The porosity of mortar with two different mix ratios is given in Table 5. According to the literature [49,50,51,52,53], pores less than 50 nm, 50–500 nm, and greater than 500 nm are defined as harmless pores, less harmful pores, and harmful pores, respectively.

Table 5 shows the pore structure of REF and SMC. Due to the existence of compressive stress, the total porosity of SMC was 10.69%, which was 32.2% less than that of REF. Moreover, the harmful pores of SMC totaled 2.38%, which was 50.7% less than that of REF. The above data showed that the combined action of MgO and steel fiber optimized the pore structure of concrete.

When steel fiber and MgO were added to concrete at the same time, MgO was used as an expansion element, and steel fiber simultaneously played a role in limiting expansion and preventing cracks. Both of them had a compound effect and produced chemical compressive stress in concrete, reduced the porosity of concrete, and optimized the pore structure, which was the key to improving the performance of SMC.

## 4. Conclusions

In this research, the effect of MC and SMC on the temperature, deformation, and strength of a bridge deck is studied. Through detailed experimental study, the following conclusions can be drawn:(1)With the increase of bridge spans, the cement content of modern concrete is increasing, which significantly increases the early shrinkage. The shrinkage of REF measured at the Xiaoqing River Bridge in 520 h reached −88.3 μ ε. If no measures are taken to reduce shrinkage, the shrinkage under the constraint of the precast slab below will produce tensile stress, which will lead to shrinkage cracks.(2)Due to the radiation of the sun, the temperature of the bridge deck is very high during the daytime, but at night the temperature will quickly drop. In this process, the bridge deck is subjected to repeated tension and compression. Because the concrete is not completely hardened in the early stage, the strength is low, and it is easy to produce temperature cracks. Therefore, in the early stage of construction, it is necessary to strengthen the maintenance of the bridge deck.(3)After adding MgO to concrete, the expansion of MgO hydration compensated the shrinkage, and the expansion of MC in 520 h was 24.9 × 10^−6^, preventing shrinkage cracks of the deck. The later expansion of MgO tended to a stable value, and there was no backward shrinkage. In addition, the strength test showed that MgO did not reduce the strength of the deck.(4)Due to the restraint of steel fibers, Ca(OH)_2_ was generated in the restricted space, resulting in pre-pressure, which improved the performance of concrete. When MgO and steel fiber were used at the same time, the expansion of SMC decreased to 9.1 × 10^−6^ in 520 h, and the strength increased by 17.5% in 21 days. The test results showed that the combined action not only prevented shrinkage cracking of the slab, but also improved the strength of the concrete. In this way, temperature cracks and shrinkage cracks could be effectively prevented.(5)The use of MgO alone had little effect on the resistance to chloride penetration of concrete, with a maximum increase of only 4.6%. On the other hand, the combined effect of MgO and steel fibers at the same time significantly improved the resistance to chloride penetration of concrete, with a maximum increase of 40.7%.(6)The combined effect of MgO and steel fiber significantly improved the pore structure of concrete, which reduced the total porosity of SMC by 32.2% and the harmful pore by 50.7%, which also explained the reason for the increase of rebound strength of SMC from the micro level.(7)Through in-situ tests at the construction site, this study clearly shows that the combined application of MgO and steel fiber can still maintain good performance in outdoor environments. Therefore, SMC, a new type of cement-based material, should be widely promoted and applied to bridge decks of long-span bridges, which has great significance in improving the durability of bridges.

## Figures and Tables

**Figure 1 materials-13-03074-f001:**
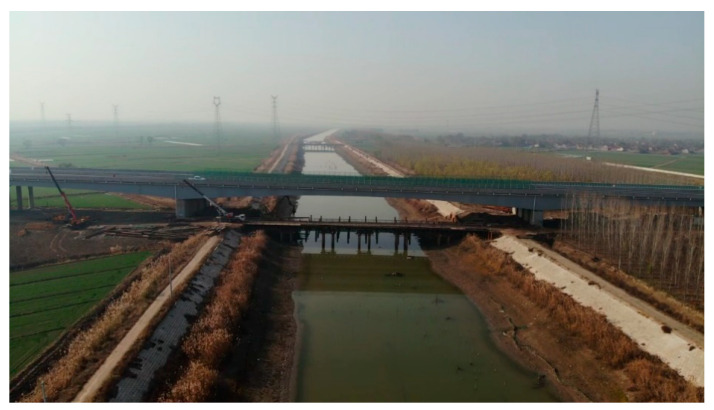
Photo of Xiaoqing River Bridge.

**Figure 2 materials-13-03074-f002:**
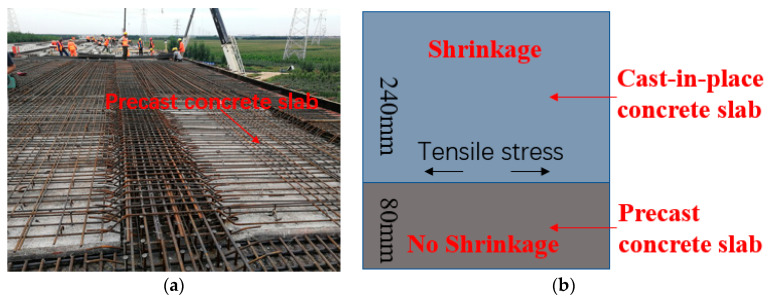
Structure of bridge deck: (**a**) construction site; (**b**) sketch map.

**Figure 3 materials-13-03074-f003:**
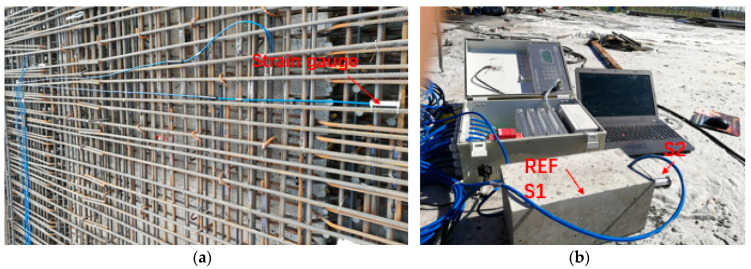
Field arrangement of strain gauge: (**a**) placement of strain gauge; (**b**) data acquisition. Where strain gauge 1 (S1) was used to measure the temperature and deformation of Ref, and S2 was used to measure the surface temperature of the deck.

**Figure 4 materials-13-03074-f004:**
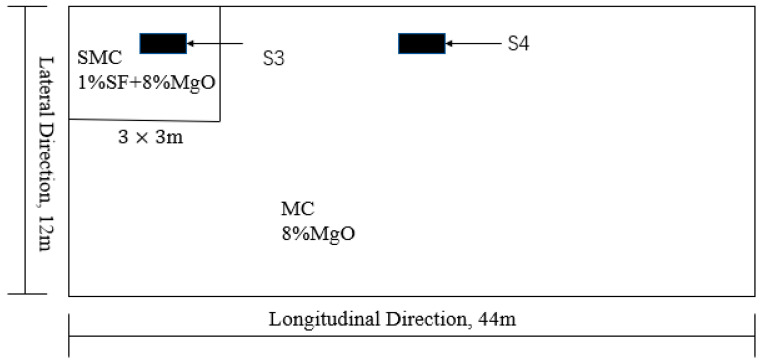
Location of testing sections. SMC: steel fiber reinforced MgO concrete; SF: steel fiber; MC: MgO concrete.

**Figure 5 materials-13-03074-f005:**
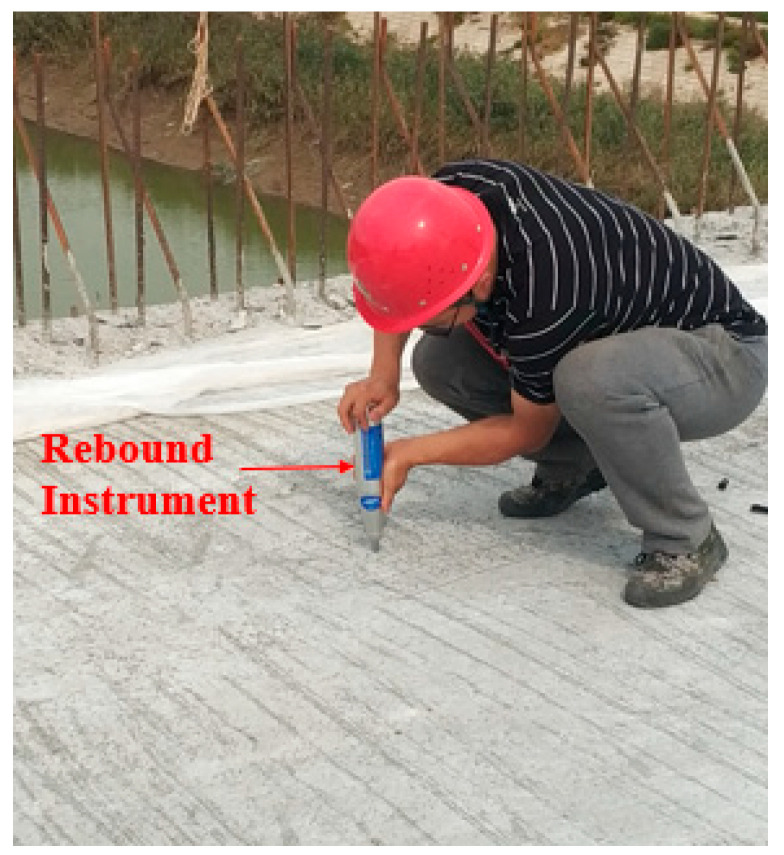
Test of the concrete strength by rebound instrument.

**Figure 6 materials-13-03074-f006:**
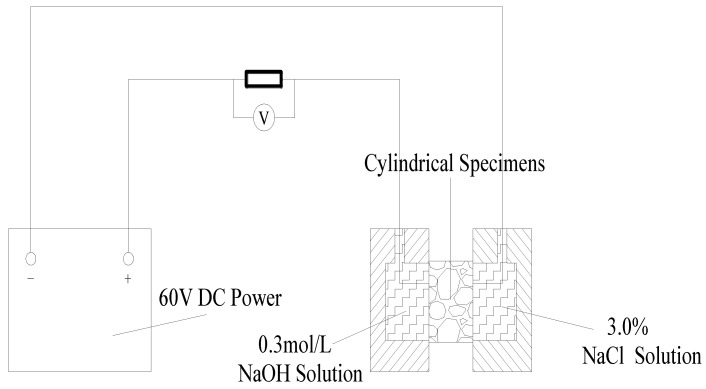
Setup for the chloride diffusion test.

**Figure 7 materials-13-03074-f007:**
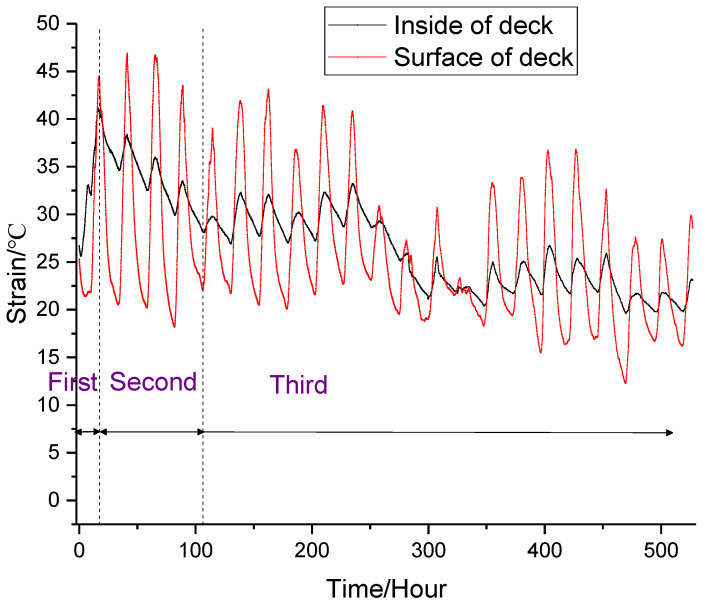
Curve of temperature variation.

**Figure 8 materials-13-03074-f008:**
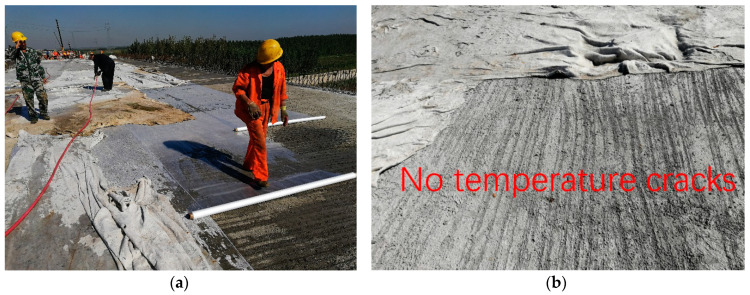
Photos of bridge deck: (**a**) curing; (**b**) 21 days after pouring.

**Figure 9 materials-13-03074-f009:**
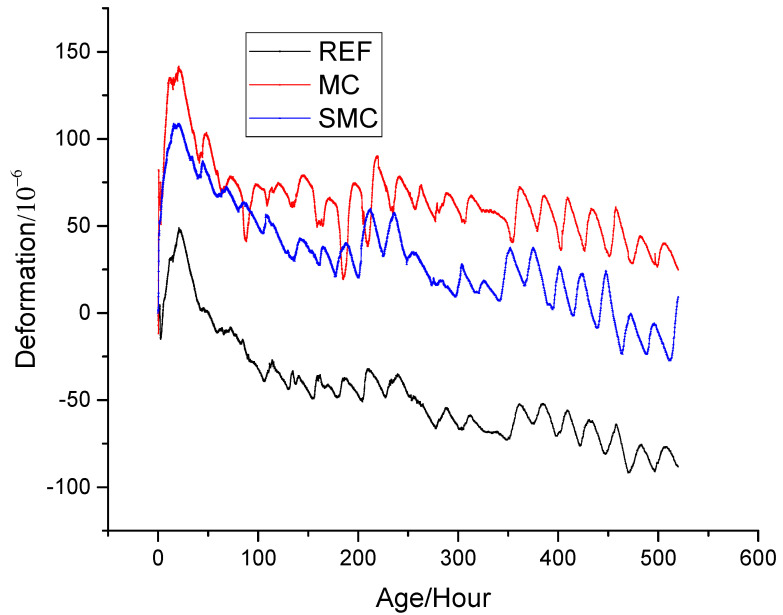
Deformation of concrete with different mix proportions.

**Figure 10 materials-13-03074-f010:**
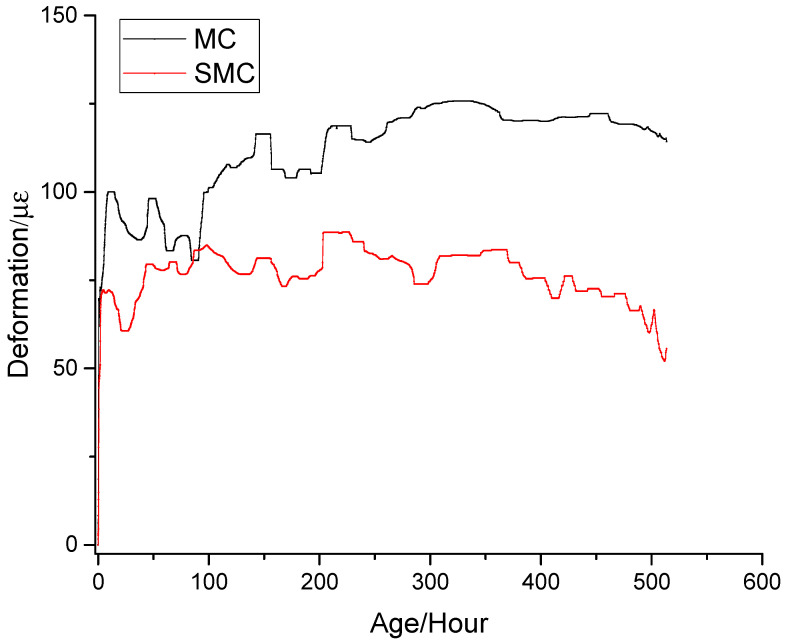
MgO expansion test.

**Figure 11 materials-13-03074-f011:**
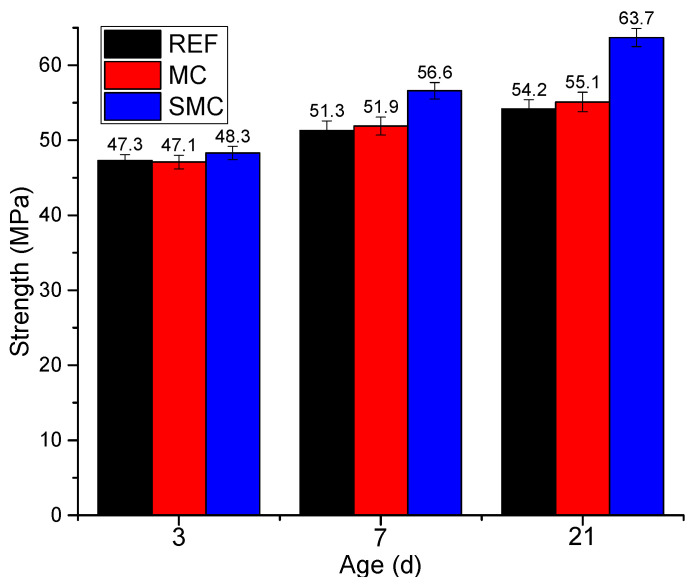
Strength variation of concrete with different proportions.

**Figure 12 materials-13-03074-f012:**
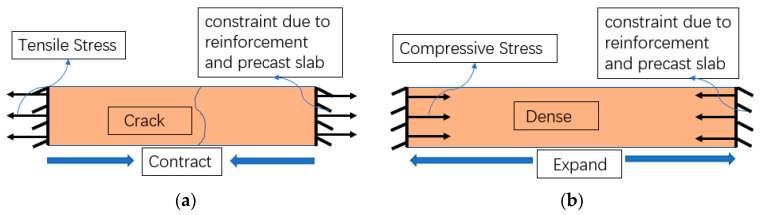
Stress produced by deformation under constraint: (**a**) REF; (**b**) MC and SMC.

**Table 1 materials-13-03074-t001:** Chemical composition of cement and fly ash.

No.	Chemical Composition/%									
	CaO	MgO	Al_2_O_3_	SiO_2_	Fe_2_O_3_	SO_3_	K_2_O	Na_2_O	Loss	Total
Cement	60.51	2.18	6.34	22.02	3.05	1.86	0.47	0.23	1.96	98.62
Fly ash	5.01	1.03	34.18	48.91	5.22	1.20	0.89	0.62	1.50	98.56

**Table 2 materials-13-03074-t002:** Basic properties of steel fiber.

Length/mm	Diameter/mm	Tensile Strength/MPa
38 (±5%)	0.58 (±5%)	520

**Table 3 materials-13-03074-t003:** Mix proportion of concrete.

No.	Composition/kg·m^−3^							
	Cement	Fly Ash	Fine Aggregate	Coarse Aggregate	Water	Water Reducer	Steel Fiber	MgO
Ref	450	50	713	1025	160	6	0	0
8%MgO(MC)	450	50	713	1025	160	7.5	0	40
1%SF + 8%MgO(SMC)	450	50	713	1025	160	7.6	78	40

**Table 4 materials-13-03074-t004:** Chloride permeability of concrete at different ages.

Type	Chloride Diffusion Coefficient (10^−9^ cm^2^/s)				Reduced Extent (%)			
	3 d	7 d	28 d	60 d	3 d	7 d	28 d	60 d
REF	15.6	15.4	10.8	8.8	0	0	0	0
MC	15.4	15.2	10.7	8.4	1.3	1.3	0.9	4.6
SMC	13.7	11.2	6.4	6.2	12.2	27.3	40.7	29.6

**Table 5 materials-13-03074-t005:** Pore structure of mortar.

No.	Total Porosity/%	Pore Distribution/%		
		0–50 nm	50–500 nm	>500 nm
REF	15.76	8.51	2.42	4.83
SMC	10.69	4.38	3.98	2.38

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
