# Peer review of "Effects of MgO Expansive Agent and Steel Fiber on Crack Resistance of a Bridge Deck"

_materials, 2020, doi:10.3390/ma13143074_

Round 1
Reviewer 1 Report
The article „Effects of MgO Expansive Agent and Steel Fiber on Crack Resistance of Bridge Deck” (Manuscript ID: materials-836762) presents the results of using MgO and MgO+steel fibres as an addition for concrete to prevent its cracking. Tests of shrinkage, variations of temperature compressive strength and resistance to chloride ions penetration were performed. In my opinion the article is interesting but it needs a serious improvement to be published in Materials journal. The main aspects which needs to be improved are:
- Introduction needs to be improved. Number of references and range of topics discussed is quite poor.
- Discussion about durability of concrete includes only the effect of addition MgO on the penetration of chloride ions. There is no discussion about risks with using MgO in cement composites and also about frost resistance of concrete. Those aspects which determines the durability of concrete are crucial for bridge decks.
- Steel fibres might affecting the results of chloride diffusion test. This amount of steel fibres in tested samples might form a shortcut between both surfaces of tested sample and received values might be wrong. This kind of test is not dedicated for reinforced concrete.
- There is no statistic for the most of presented results or it is poor.
Author Response
Dear Professor
For your comments, I have explained one by one in the attachment.

Reviewer 2 Report
In my opinion, the document has no structure for a scientific paper to be published in an indexed journal. The theme is interesting and if the work had a different structure it would present an interesting publication potential. However what we have in this paper is:
- Only 3 mixes under analysis;
- An experimental campaign with few tests and organized incoherently (for example: Strength Test made using the concrete rebound hammer exclusively - this cannot be done like this!);
- Results analysis is simply nonexistent. The authors limit themselves to presenting the results obtained; there is no attempt to analyze the very few results obtained.
- There is no attempt to benchmark the results; the results obtained are not correlated with each other or with the values of other authors or even with reference values.
In my opinion, although the authors added the results of 2 more tests, the paper has a fundamental problem regarding its structure, how it is organized and the content itself makes it impossible to use it in an indexed journal.
Author Response

(The authors gave the same response as above.)

Reviewer 3 Report
Reviewed paper present with no doubt original research referring to the vital problem of shrinkage cracking in large structures.
Researching team proposed well known, but a relatively new solution to this problem MgO addition to concrete and MgO together with dispersed steel fibre reinforcement.
The main quality of the research comes from the fact that it was realized in situ on a large bridge structure.
The observations made during the tests made on the real-scale structure in real conditions will undoubtedly be valuable for readers and other researchers.
The Introduction does not exhaust, in my opinion, the issue of the shrinkage in large concrete structures and the use of MgO and dispersed fibre reinforcement to reduce this unwanted effect. The literature related to this problem is very rich and widely discussed, referring to many aspects. Formally, Authors of this study quote 20 papers in the Introduction; however, it does not bring satisfying information to the reader trying to understand the reasons standing behind described here research. Authors do not also refer sufficiently to their previous research, which could allow understanding the reasons for applying described in this paper solution to the Xiaoqing River Bridge.
One of the main questions, which rises here is why Authors decided to apply the unverified solution to the responsible structure as the bridge in the strategic highway network? Authors refer to their previous test [20], but they seem to be the initial test.
What are the requirements of Chinese Standard to permit to use new solutions in real life? Does the research meet those requirements?
In my opinion, there are many missing points in initial tests (at least not shown here), not related directly with the shrinkage but for example with durability, fatigue and rheology (i.e. creeping during long term load application).
Regarding the research in detail, many data are missing, or it is not clear, in my opinion:
- General scheme of the bridge with the scheme of concreted in situ deck segments (with dimensions) would be important for the readers. It is unclear where the MC and MSC segments are situated on the bridge. How far is it from bearings? Is the segment of 44m surrounded by regular concrete? Is the MC segment cast with no expansion joints? Where exactly are the testing sections, how large are they? Fig. 4 does not explain it. All this information and data should be clearly shown in the text, best on the draw.
- Authors state that prefabricated bearing elements are shrinkage free. Authors base on the information, that those elements were built in month before, but there is no record on the total age of those elements. How do we know, that the shrinkage has ended if, as we know, in this type of elements it may take min. 90 days to stabilize the shrinkage.
- Referring to the time of the testing, Authors present the results for 21 days only, while in this type of element with this height, the shrinkage and other rheological behaviours may demonstrate double effects (measuring strains for example) after 180 days and in fact it may be treat as ended after 720 days. Do the Authors dispose of further measurements and observations?
- Was there any bonding layer used between “old” and “new” layers of concrete? If so, which one? If not – was it based on any analyze? May the Authors comment on that.
v. 35 – “under free condition” – unclear
v. 89 – 90 – “concrete of bridge deck is able to maintain good field performance under the field harsh environment” – what does it mean “good field performance” and “harsh environment” – it is not clear, especially referring to the fact, that the tests were conducted during 21 days period only
Formal, editorial issues:
Numerous missing determiners. The English language should be double-checked. There are some minor flaws, errors and repetitions in the text.
Figures are described with different shape and size of fonts; it would be better to unify them. Fig. 6 especially looks different than the others.
Fig. 2, 3 descriptions placed in the image are not readable.
Equations are written with larger font size than the rest of the text.
v. 20 Field – should be field (or maybe in situ).
In general, the text is interesting and brings interesting observations. Most precious element is a large scale, in situ test. Still, to understand the mechanism and develop a proposed method or use it in other applications, there are many missing information and data.
It would be excellent if the researching team might add information to the text or comment on the addressed above issues.
Regarding all the listed above problems, the conclusions should be revised and take those issues into consideration.
Author Response

(The authors gave the same response as above.)

Reviewer 4 Report
The present manuscript is on the practical application of MgO for the concrete to reduce the shrinkage cracking.
Most of all, the present manuscript has a novelty and scientific values, as there are limited reports on the application of MgO concretes.
Minor comments are bellows
- The mix proportion in Table 3 should be modified. Even the 8 % of MgO was applied in the mixtures, there were no changes on the unit weights of other constituents of REF. As MgO contents increased, other contents should be decreased. However, the still fiber contents was just 1 vol.%, so it could be ignored.
- In Fig. 5, it seems like that the surface of concrete was not considered for the rebound test. If the surface was not flattened, the reliability of the test results become very low.
Author Response

(The authors gave the same response as above.)

Round 2
Reviewer 1 Report
In my opinion the paper after revision has not been improved enough to be published in the Materials Journal with an Impact Factor. Most of points suggested in previous revision are still missing. New parts of text added after revision has new flaws as described below. I suggest that the article should be rejected.
line 21-23: The Authors declares that "The research on the performance of concrete in the test section showed that MgO and steel fiber were safe for the bridge deck..." It is not justified without showing any results or even references that might presents the long-term durability of this kind of concrete.
line 45: The Authors suggested that high shrinkage is due to high compressive strength of concrete. High shrinkage is related with high amount of clinker in concrete which in fact gives the high compressive strength.
line 46: "...contact with air, which will cause greater chemical shrinkage..." please describe what does the Authors mean by the chemical shrinkage caused by air.
In permeability tests (section 2.2.3) Authors are measuring actually the charge transfer. When the concrete has no metal particles in it (such as steel fibers) the charge transfer might be correlated with the diffusion of chloride ions. But in this case some the chloride ions are migrating to the steel fibers and than the charge passes easily by the electrons and through the fibre. This shows that results received from the experiment might be false especially when comparing results for samples with and without steel fibers. In the test method ASTM C1202 there is no information that it is suitable for testing samples with metal particles.
Author Response
Thank you for your comments. I'm sorry that the language organization and expression in the previous part of our introduction are confused, and some important points are not provided with corresponding references. This time, we have made great adjustments to the introduction. In the attachment, we explain your comments one by one.
This paper involves two subjects: Bridge Engineering and Materials Science, so I didn't integrate the knowledge of the two subjects well in the introduction part before. The language is quite confusing. I'm really sorry to bring you reading difficulties. As a graduate student, thank you for your patience in guiding my thesis. In the future work and research, I will remember your teaching.

Reviewer 2 Report
the authors answered satisfactorily to all questions.
Author Response
Thank you very much for your help in the paper. Your suggestion has greatly improved the quality of the paper
Reviewer 3 Report
Thank you for all of your comments in the Cover Letter, they are very interesting. It is a pity, that described in the Cover Letter knowledge is not highlighted so well in the text of your paper. It would certainly give better background for your work.
Text still needs extensive proofreading, there are many, minor, but still, language mistakes.
Wish you and your team many successes in the future, looking forward to read your further papers on interesting realisations.
Author Response
Your suggestion has greatly improved the quality of the paper. We have revised some language mistakes in the paper again. Thank you very much.